

# Spatio-temporal evolution of glacial lakes in the Tibetan Plateau over the past 30 years

Xiangyang Dou[1], Xuanmei Fan[1*], Ali.P Yunus[2], Junlin Xiong[1], Ran Tang[3], Xin Wang[1], Qiang Xu[1]

[1]State Key Laboratory of Geohazard Prevention and Geoenvironment Protection, Chengdu University of Technology, 610059, Chengdu, China

[2]Department of Earth and Environmental Sciences, Indian Institute of Science Education and Research Mohali, Mohali, Punjab, 140306, India

[3]School of Architecture and Civil Engineering,  Chengdu University, 310106, Chengdu, China

*Correspondence to*: Xuanmei Fan (fxm_cdut@qq.com)

**Abstract.** As the Third Pole of the Earth and the Water Tower of Asia, Tibetan Plateau (TP) nurtures large numbers of glacial lakes, which are sensitive to global climate change. These lakes modulate the freshwater ecosystem in the region, but concurrently pose severe threats to the valley population by means of sudden glacial lake outbursts and consequent floods (GLOFs). Lack of high-resolution multi-temporal inventory of glacial lakes in TP hampers a better understanding and prediction of the future trend and risk of glacial lakes. Here, we created a multi-temporal inventory of glacial lakes in TP using 30 years record of 42833 satellite images (1990-2019), and discussed their characteristics and spatio-temporal evolution over the years. Results showed that their number and area had increased by 3285 and 258.82 km$^2$ in the last 3 decades, respectively. We noticed that different regions of TP exhibited varying change rates in glacial lake size, most regions show a trend of expansion and increase in glacial lakes, while some regions show a trend of decreasing such as the western Pamir and the eastern Hindu Kush. The mapping uncertainty is about 17.5%, lower than other available datasets, thus making our inventory reliable for the spatio-temporal evolution analysis of glacial lakes in TP. Our lake inventory data are freely available at https://doi.org/10.5281/zenodo.5574289 (Dou et al., 2021), it can help to study climate change-glacier-glacial lake-GLOF interactions in the Third Pole and serve input to various hydro-climatic studies.

## 1 Introduction

The "Third Pole of the Earth" (Qiu, 2008), Tibetan Plateau (TP), contains the most significant number and area of glaciers outside the Antarctic and the Arctic (Yao et al., 2012). With the aggravated climate change in the anthropocene, the retreat and loss of glacier mass increased in many parts of the TP (Bolch et al., 2012; Brun et al., 2017; Gardner et al., 2013; Hugonnet et al., 2021; Kääb et al., 2015; Shean et al., 2020; Zemp et al., 2019). This trend intensified in the last few decades, with an accelerated rate (−0.18 to −0.7 m w.e. yr$^{-1}$) since the mid-1990s (Bolch et al., 2012; Brun et al., 2017). The melting snow and ice present a chance for the development of glacial lakes. Many glacial lakes form in the low-lying land, such as



depressions and troughs, and gradually expand with precipitation or glacial retreat and melt supply (Clague and Evans, 2000; Mool et al., 2001; Song et al., 2016; Wang et al., 2020).

Glacial lakes are both temporary reservoirs of glacial meltwater and potential sources of flooding (Wang et al., 2020). Although the frequency of GLOFs in the TP region has not increased significantly in recent years (Veh et al., 2019), the potential risk posed by glacial lake expansion remains. Unanticipated GLOFs bring potential dangers to downstream

communities and their infrastructure, as well as affecting the regional ecological environment (Bolch et al., 2012; Haeberli et al., 2017; Huggel et al., 2002). In addition, these glacial lakes play a great important role in the ecosystem dynamics and hydrological cycle of the region. A growing number of scientific research and policy concerns are therefore recognized in TP dealing with the aforementioned two issues related to glacial lakes (Woolway et al., 2020; Yang et al., 2011; Zhang et al., 2020a).

Since the 1980s, scholars have been continuously studying the glacial lakes in TP and mapped them based on different methods and means. Thanks to the development of various remote-sensing technology and the massive leap in computing power of computers, large-scale regional studies have been increasingly applied in the field of geological disasters and in the cryosphere environment. In particular, Landsat satellites, with their free access, high-revisit ability (16 days) and high spatial resolution, have become the preferred data source for long-term monitoring and research in most regions (Irons et al., 2012).

At the same time, the improvement of cloud-computing power, such as the application of Google Earth Engine (GEE), has dramatically improved the efficiency of regional spatial analysis (Gorelick et al., 2017). All these have greatly improved the accuracy of automatic and semi-automatic glacier lake boundary vectorization. Compared with manual visual interpretation of lake mapping, automated techniques are more efficient and have been widely used in lake extraction studies (He et al., 2021; Zhao et al., 2018). Nevertheless, to reduce the systematic error in automation, some amount of manual correction is

still indispensable (Wang et al., 2020).

To our best knowledge, about 30 glacial lake datasets or reports have been published in the TP area, each using different extraction methods and data sources (see the Supplement Table S1 of Wang et al., 2020). Most of them adopted the normalized difference water index (NDWI) to extract the lake boundaries (Ashraf et al., 2014; Bolch et al., 2008; Bolch et al., 2011; Chen et al., 2021; Gardelle et al., 2011; Jain et al., 2015; Khadka et al., 2018; Mool et al., 2001; Nie et al., 2013; Nie et

al., 2017; Prakash and Nagarajan, 2018; Shrestha et al., 2017; Shukla et al., 2018; Wang et al., 2016; Wang et al., 2017b; Wang et al., 2013b; Worni et al., 2013), while some others used manual interpretation and auxiliary technologies, such as "Global-local" iterative scheme, band ratio threshold condition, integrated nonlocal active contour approach and machine learning models etc. (Li et al., 2011; Liu et al., 1988; Maharjan et al., 2018; Petrov et al., 2017; Raj and Kumar, 2016; Senese et al., 2018; Song et al., 2016; Song et al., 2017; Veh et al., 2018; Wang et al., 2015; Wang et al., 2013a; Zhang et al.,

2015; Zhang et al., 2018a).

Despite the large volume of studies, there was no unified standard about the minimum threshold area applied to extract the glacial lakes; different studies adapted different threshold areas in literature. For example, Salerno et al. (2012) used 0.001 km$^2$ as the minimum threshold area to research the glacial lakes in the Mount Everest region. Wang et al. (2013b) mapped



the glacial lakes with an area of >0.002 km$^2$ in Tian Shan and central Asia. Zhang et al. (2015), on the other hand made systematic research of glacial lakes larger than 0.0027 km$^2$ in TP; Gardelle et al. (2011) and Luo et al. (2020) used 0.0036 km$^2$ as the minimum area to study the climatic response of glacial lakes in the Hindu Kush Himalaya mountain range and western Nyainqêntanglha range separately; Li and Sheng (2012) presents an automated scheme for glacial lake dynamic mapping in the Himalayas with the minimum area of 0.0045 km$^2$. Lately, Wang et al. (2020) updated the minimum threshold area as 0.0054 km$^2$ to study the glacial lake changes in TP; and Nie et al. (2017) selected 0.0081 km$^2$. Li et al. (2020) and Worni et al. (2013) individually produced glacial lake inventory in China-Pakistan Economic Corridor (CPEC) and Indian Himalayas with the minimum threshold area as 0.01 km$^2$. Some larger values of minimum threshold areas such as 0.02 km$^2$ and 0.1 km$^2$ were also applied to analyze the GLOFs (Allen et al., 2019; Bajracharya and Mool, 2009; Bolch et al., 2011; Wang et al., 2017a; Wang et al., 2011; Zhang et al., 2019). According to the properties of the satellite images, the minimum pixel length used to extract the glacial lake is 30 m, which means that some glacial lakes cannot ideally occupy a complete number of pure pixels but are more likely to be partially surrounded by 1-8 mixed water body pixels (Wang et al., 2020). Studies have shown that applying a smaller minimum threshold area under the same spatial resolution will significantly increase the total number of glacial lakes, but the general area of the lakes will not change significantly (Nie et al., 2017). Nevertheless, choosing a minimum threshold area that is too small will lead to substantial uncertainty (see Sect. 4.1) and significantly increase the workload of meaningless cross-validation and manual correction (Salerno et al., 2012), resulting in a negative impact on extracting glacial lake in TP.

Aforementioned literature demonstrated systematic studies on the glacial lakes, but most of them focused a specific region rather than the whole of Tibetan Plateau. While some works covered the entire range, there is still a lack of multi-temporal, long-term monitoring and comprehensive analysis of the glacial lakes over the entire TP region (Aggarwal et al., 2017; Chen et al., 2021; Chen et al., 2007; Wang et al., 2020). In the time of increased warming trends, it is of great significance to study the change trends of glacial lakes in TP over a long time period. In order to address the above problems of incomplete spatial and temporal coverage of glacial lake data on the TP, this study mapped an updated inventory of glacial lake covering the entire TP including three periods of data, aiming to solve the above problems and provide a data base for cryosphere studies.

## 2 Study area

Tibetan Plateau, also called "the Roof of the World" (Qiu, 2008; Zhang et al., 2020b), has a mean elevation of ~4000m a.s.l., with higher elevation in the west and lower in the east. The total area of TP is ~3×10$^6$ km$^2$, most of which is in China, with other parts in India, Pakistan, Afghanistan, Tajikistan, Kyrgyzstan, Nepal, Bhutan and Myanmar (Zhang et al., 2020b).

Many high mountains surrounded TP, including Pamirs and Hindu Kush Mountains in the west, Altun Mountains, Kunlun Mountains and Qilian Mountains in the north, the Himalayas in the south, and Hengduan Mountains in the east. (Figure 1). Among these mountains, only the Hengduan Mountains are north-south range, the rest of the mountains aligned in generally east-west orientation (Chen et al., 2021; Zhang et al., 2020b).



As "the Water Tower of Asia" (Barnett et al., 2005; Immerzeel et al., 2010), TP is the source of several great rivers, including the Amu Darya, Indus, Ganges, Yangtze, Mekong, Yellow, Salween, Brahmaputra and Irrawaddy. These rivers, which pass through many countries in Asia, especially China, India and Southeast Asia, play an irreplaceable hydrological role in providing water for domestic and industrial use to billions of people downstream (Immerzeel and Bierkens, 2010;

Immerzeel et al., 2020).

Meteorological data on the TP have been continuously tracked by establishing weather stations, with the earliest data going back as far as ~1930s (Liu and Chen, 2000). Climatic data show a higher rate of warming in 1980-2018 compared to 1961-2015. Since 1960s, the TP has warmed at twice the rate of the global average, and precipitation increased faster in 1998-2018 than in 1980-1997 (Zhang et al., 2020b), implying a warming and wetting trend in the past decades, which led to a

glacier retreat and a significant impact on the hydrological changes (Kang et al., 2010; Kuang and Jiao, 2016; Li et al., 2010; Liu and Chen, 2000; Xu et al., 2008).

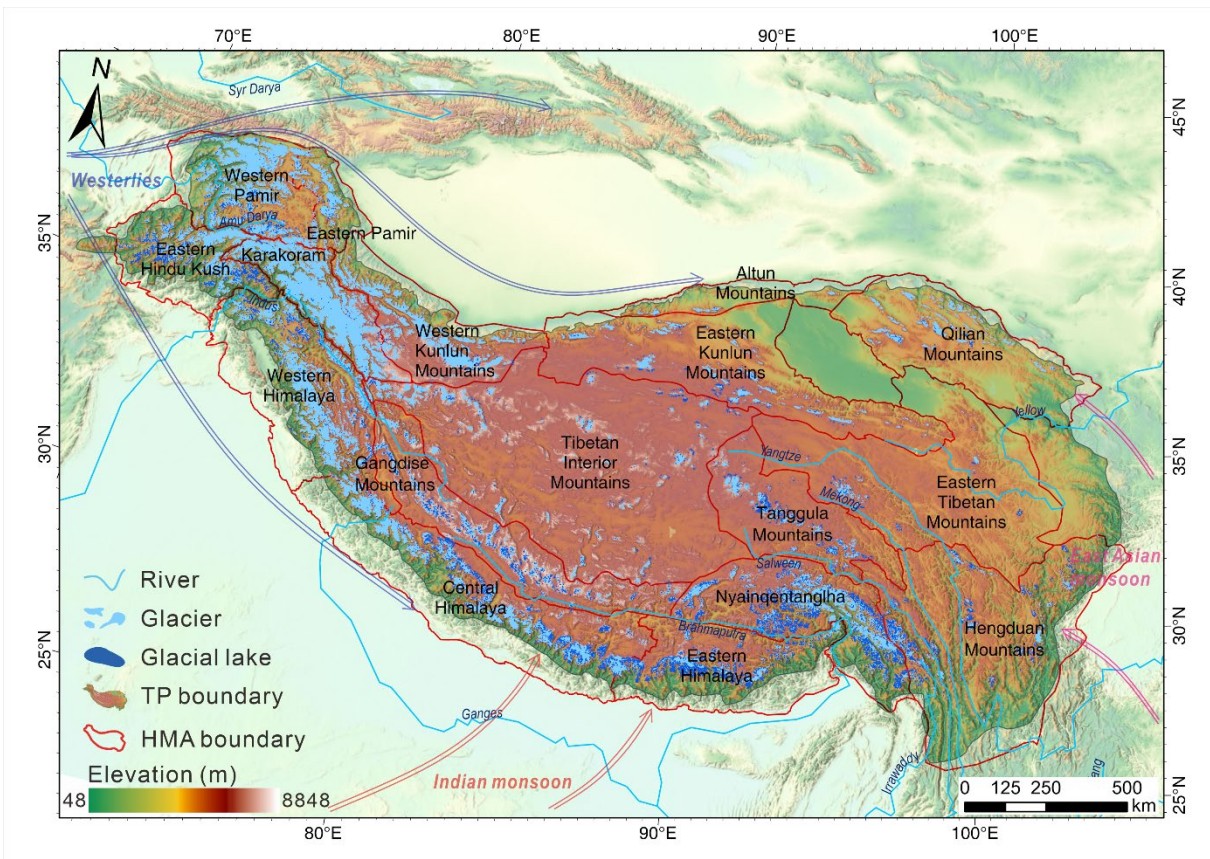

**Figure 1. Distribution of glaciers, glacial lakes and major rivers on the Tibetan Plateau (TP). The TP was divided into 17 mountains (http://geo.uzh.ch/~tbolch/data/regions_hma_v03.zip). The large-scale atmospheric circulations are also included. The**

**terrain basemap is sourced from ESRI.**



## 3 Data and Methods

We applied a two-step method to construct the Tibetan Plateau Glacial Lake database (TPGL) from 1990 to 2019. A total of 42833 (12224, 14670, and 15939 for the period 1990-1999, 2000-2012, and 2013-2019 respectively) Landsat Surface Reflectance (SR) images were preprocessed on Google Earth Engine (GEE), which has mighty computing power based on
cloud computing to cope with complex and large workloads (Gorelick et al., 2017; Kumar and Mutanga, 2018). Subsequent processing was handled by © ArcGIS Pro and © ENVI software, including manual cross-checking and correction by image interpretors. The general workflow of method is shown in Figure 2.

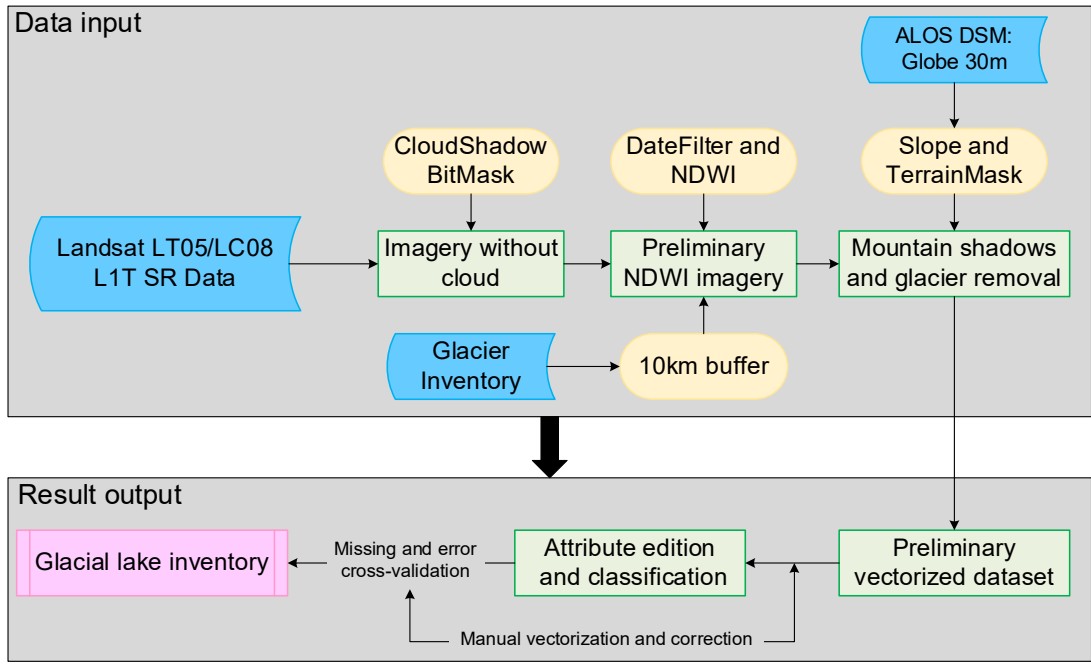

**Figure 2. Flowchart of the glacial lake automatic extraction and mapping workflow.**

**3.1 Data**

Because of data strip issues in Landsat 7 ETM+ caused by sensor failure, we mainly used Landsat 5 Thematic Mapper (TM) and Landsat 8 Operational Land Imager (OLI) for image processing. Typically, period of July to September months corresponding to summer months is considered to be the ideal time for glacial lake mapping. During this period, the coverage of snow and ice is minimal, while the glacial lake area usually reaches its maximum. Under the action of glacial
meltwater and precipitation, the glacial lake area does not produce large fluctuations (Nie et al., 2017; Zhang et al., 2015). In the absence of cloud-free data, images from the nearest time period can be selected as a substitute (Chen et al., 2021; Nie et al., 2017). Although the melting rate of snow and ice peaks in July and August with increase in surface temperature (Ding et al., 2018; Gardelle et al., 2013), we chose a conservative range in this study, i.e., from July to November with the consideration of obtaining more available cloud-free remote sensing images.





Since the original images may contain clouds and mountain shadows, essential preprocessing was carried out in GEE to
       mask clouds and cloud shadows (Beckschäfer, 2017; Li et al., 2018; Skakun et al., 2019; Zhu and Woodcock, 2012). Here
       we used the method of "cloudBitMask" and "cloudShadowBitMask" with a threshold of 1-5 and 1-3 to mask the cloud and
       its shadow in GEE (Gomez-Chova et al., 2017; Mateo-Garcia et al., 2018). Then ALOS AW3D-30 m digital elevation model
       (DEM) was employed to eliminate the effects of slope and topographic shadows. We set 7° slope as the masking threshold to
eliminate some pseudo glacial lakes (Li and Sheng, 2012; Quincey et al., 2007). Since the production time of ALOS DSM
       data was not consistent with the acquisition time of Landsat SR images used for glacial lakes mapping, the resulting slope
       and terrain may not match the actual terrain completely, leading to minor errors in masking glacial lakes. These errors were
       corrected as much as possible in the subsequent cross-validation and manual correction steps (see Section. 3.4).

### 3.2 Glacial lake mapping

The distance between the glacial lake and its nearest glacier terminus is one of the criteria of identifying a glacial lake. In
       previous studies, several distance values, such as 2, 3, 5, 10, and 20 km, were used as the maximum threshold value for
       glacial lake identification (Petrov et al., 2017; Veh et al., 2018; Wang et al., 2013b; Zhang et al., 2015). Nie et al. (2017) and
       Zhang et al. (2015) attributed a distance of 10 km from the nearest glacier terminus as a reasonable threshold. To ensure the
       consistency and comparability of the data, we selected the buffer of 10 km distance as the spatial distribution range of the
glacial lakes in this study. After comparing the published glacier inventories covering the TP, including the Global Land Ice
       Measurements from Space (GLIMS) glacier database (Raup et al., 2007), the Randolph Glacier Inventory (RGI) (Arendt et
       al., 2017; Pfeffer et al., 2014), the Glacier Area Mapping for Discharge from the Asian Mountains (GAMDAM) glacier
       inventory (Nuimura et al., 2015; Sakai, 2019), and the First and Second Chinese Glacier Inventory (CGI) (only covered the
       Tibet Autonomous Region of China) (Guo et al., 2015; Shi et al., 2009), and many others (e.g., Jiang et al., 2018; Paul et al.,
2013; Raup et al., 2013; Smith et al., 2015), we believe that GAMDAM is superior in terms of data accuracy (based on the
       fact that it is done entirely by manual mapping) and updated time (2018) (He and Zhou, 2022), and therefore use GAMDAM
       to determine the extent of glacial lake distribution.

       Different indices were proposed and employed to extract water bodies based on remote sensing imagery, such as the
       normalized difference water index (NDWI) and the modified normalized difference water index (MNDWI). These two
indices utilize green and NIR or SWIR bands to extract water pixels (the relevant formulae can be found in references, e.g.,
       Li et al., 2016; Mcfeeters, 2007; Xu, 2007; Zhai et al., 2015; Zhang et al., 2018b). Before properly extracting the lakes of the
       whole TP, we randomly selected several areas and employed MNDWI and NDWI to extract the lake pixels as a test case.
       Based on the test results, NDWI has a better extraction effect and accuracy than MNDWI, so NDWI was finally used in this
       study to extract the lake outlines automatically.

Because Landsat 5 TM and Landsat 8 OLI images have different band properties for NDWI implementation, a universal
       threshold cannot be simply applied to automatic extraction of glacial lakes in different periods. In this study, a dynamic
       range from -0.1 to 0.2 was determined based on the careful consideration of the thresholds used by predecessors (Du et al.,



2014; Liu et al., 2016). The best threshold is selected based on the degree of matching by manually checking the processed images and the extraction results of multiple attempts with different thresholds. After GEE implementation, we carried out
further processing of NDWI output of Landsat images by using the Majority Analysis and Clump Classification function of © ENVI. Finally, the raster dataset was converted into vector files in © ArcGIS Pro, achieving the outlines of glacial lakes. Thus, based on the uncertainty and spatial resolution, this study set 0.0081 km$^2$ (3×3 pixels) as the minimum threshold area according to experience and multiple attempts in different regions of TP. Further cross-validation, manual vectorization, and correction were all based on this precondition.

## 3.3 Glacial lake classification

Based on the origin and location of glacial lakes (Chen et al., 2021; Nie et al., 2017; Rick et al., 2022; Vilímek et al., 2013; Wang et al., 2020; Yao et al., 2018), we classified the mapped lakes into the following four types (as shown in Figure 3):

(i)     Proglacial lakes (PGL): the lakes are connected to the glacier and located in the front of the glacier (glacier tongue), usually dammed by moraines. some of them are fed directly by glaciers (Carrivick and Tweed, 2013; Yao et al., 2018);

(ii)    Supraglacial lakes (SGL): the lakes developed on the glacier surface, surrounded in the whole or in part by glaciers (Benn et al., 2017; Reynolds, 2000);

(iii)   Ice-marginal lakes (IML): usually located on the side of the glacier tongue and dammed by lateral moraines. It is commonly found in areas such as Alaska and has only a small distribution in TP (Armstrong and Anderson, 2020; Capps et al., 2011);

(iv)    Unconnected glacial lakes (UGL): the lakes are not directly connected to the parent glaciers, but they may have evolved from a proglacial lake or supraglacial lake as glaciers retreat (Chen et al., 2021). Some researchers further categorized them into glacier-fed and non-glacier-fed lakes (Khadka et al., 2018; Zhang et al., 2015).

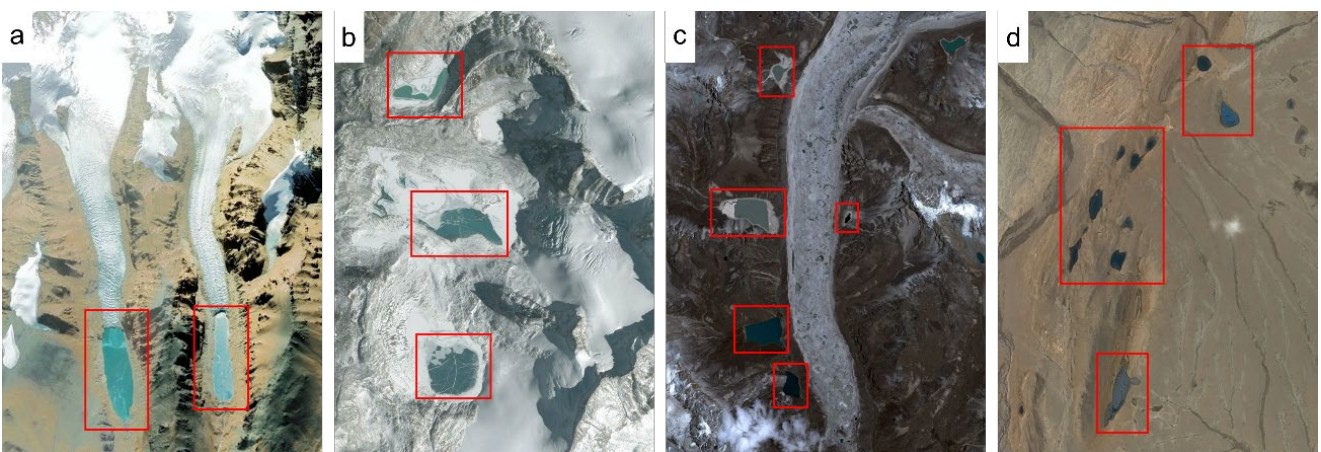

**Figure 3. Examples of each type of glacial lakes (in red rectangle) in TP: (a) proglacial lakes; (b) supraglacial lakes; (c) ice-marginal lakes; (d) unconnected glacial lakes. Background imageries were obtained from © Google Earth Pro.**



### 3.4 Estimation method of mapping uncertainties

To further improve the accuracy and reliability of the glacial lake inventory, manual vectorization was carried out by trained interpreters. Necessary corrections for each glacial lake were made and their types were identified. Although this work

involves much time and human resources, it can significantly improve the data quality.

The spatial resolution of satellite images affects the mapping uncertainty (Chen et al., 2021; Fujita et al., 2009; Salerno et al., 2012; Wang et al., 2020), and the subjectivity and experience of interpreters will also lead to errors. Considering the spatial resolution of images used in this study (30 m), and for a better cross-validation with other inventories using this threshold, Eq. (2) from Krumwiede et al. (2014) was applied to analyze the uncertainty estimation of glacial lake area delineation.

$$A_{er} = 100 \cdot (n^{1/2} \cdot m)\big/ A_{gl} \,, \tag{2}$$

where $A_{er}$ is the percentage error of area determinations, which is proportional to satellite sensor resolution. $n$ is the pixel number occupied by glacial lake boundary, in which it can be represented by the ratio of perimeter length to spatial resolution. $m$ is the area of a pixel in the image (m², e.g., 900 m² for a pixel in the Landsat imagery). $A_{gl}$ is the lake area (m²), and the factor 100 is the coefficient to convert to a percentage.

**4 Results**

### 4.1 The uncertainty of glacial lake area

The number of glacial lakes extracted in three periods (1990-1999, 2000-2012, and 2013-2019) is 19183, 20655 and 22468, and the total area is 1509.17 km², 1637.01 km² and 1767.99 km², respectively. Taking ±1 pixel (30 meters) as the uncertainty of glacier lake boundaries, we calculated the systemic errors of all glacial lakes in TP with the three periods (as shown in

Figure 4). The average uncertainty for all glacial lakes is 17.5%, with a standard deviation of 9.91% and overall uncertainty in the range of 0.2%-50%. Due to improved Landsat 8 OLI image quality, the average uncertainty for 2013-2019 was found to be the lowest. As can be seen from Eq. (2) and Figure 4, the smaller the area of a glacial lake, the higher the area uncertainty, which indicates that the area uncertainty is directly related to the size and shape of a glacial lake (Chen et al., 2021). Lower minimal area threshold for mapping will largely increase the overall area uncertainty, therefore we chose

0.0081 km² (3×3 pixels) as the area threshold.



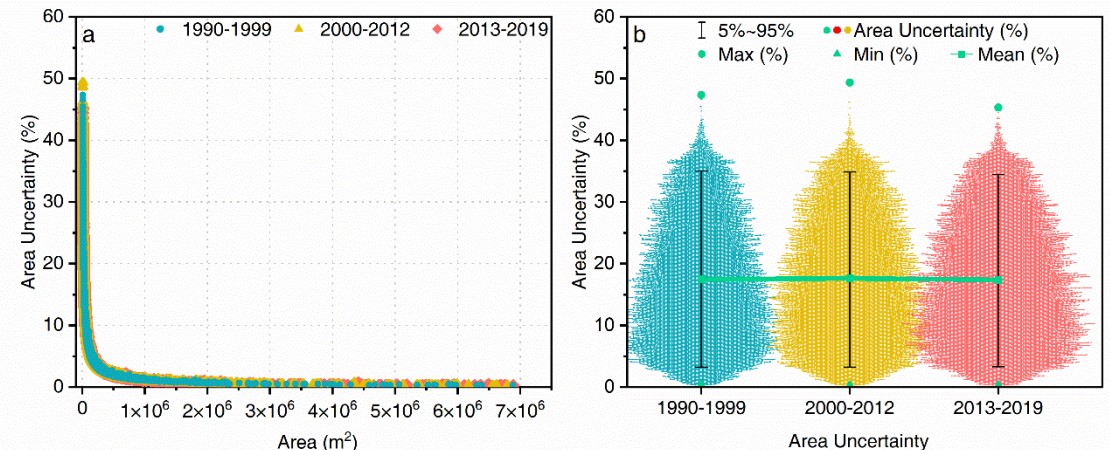

**Figure 4. (a) Relationships of relative area uncertainty of glacial lakes in TP; (b) normal distribution of uncertainty of glacial lake area, the swarm plots for each time period represent the uncertainty distribution of all glacial lakes in that period.**

### 4.2 Temporal and spatial distribution of glacial lakes

From the first period (1990-1999) to the last period (2013-2019), the overall numbers and cumulative areas of glacial lakes are all increased, in which the UGL and PGL make the largest contribution to the lake area increase under the effect of glacial retreat (see Figure 5a). Although some glacial lakes are contracting or even disappearing, six glacial lakes with significantly expanded or contracted changes were selected as examples to show their detailed outline changes in Figure 6. The area of glacier lakes distributed at elevation from 4000 m to 5300 m above sea level (a.s.l.) increased most apparently,

while the number increased most sharply for the lakes with elevation from 4000 m to 5900 m (see Figure 5b and 5c). Between 5300 m and 5900 m, the number of glacial lakes has increased as well but the area expansion was smaller compated to the range of 4000m to 5300m (exception of a few glacial lakes with a dramatic increase in area, e.g., Figure 6), indicating many small glacial lakes are formed within this elevation range. The increase of glacial lake number at higher elevations (above than 4000m a.s.l.), as well as the number of ultra-small glacial lakes (the area is between 0.001-0.0081 km$^2$) that have

been studied (Salerno et al., 2012) but not considered in this paper, suggests that glaciers start retreating at higher elevations (Chen et al., 2021; Nie et al., 2017).

Different glacial lake types have distinct characteristics in altitude distribution, and their numbers and areal distribution show the same increasing or decreasing trend with the increase of altitude. Most of the glacial lakes are distributed within the range of 3000 m to 6000 m a.s.l., in which unconnected glacial lakes (UGL) and proglacial lakes (PGL) are the dominant

cases. Whereas supraglacial lakes (SGL) and Ice-marginal lakes (IML) are small in number, and their changing trends were not prominent (e.g., in the last period of 2013-2019, see Figure 7).





**Figure 5. (a) Numbers and areas with three periods of all glacial lakes and PGL, UGL; (b) altitudinal distribution (100 m bin sizes) of lake areas; (c) altitudinal distribution (100 m bin sizes) of lake numbers.**





**Figure 6. Examples of glacial lake expansion and contraction, the location was the approximate latitude and longitude and the height of the lake's center. Background imageries were obtained from Landsat satellite images preprocessed by GEE (for automatic extraction, see Section. 3.1).**



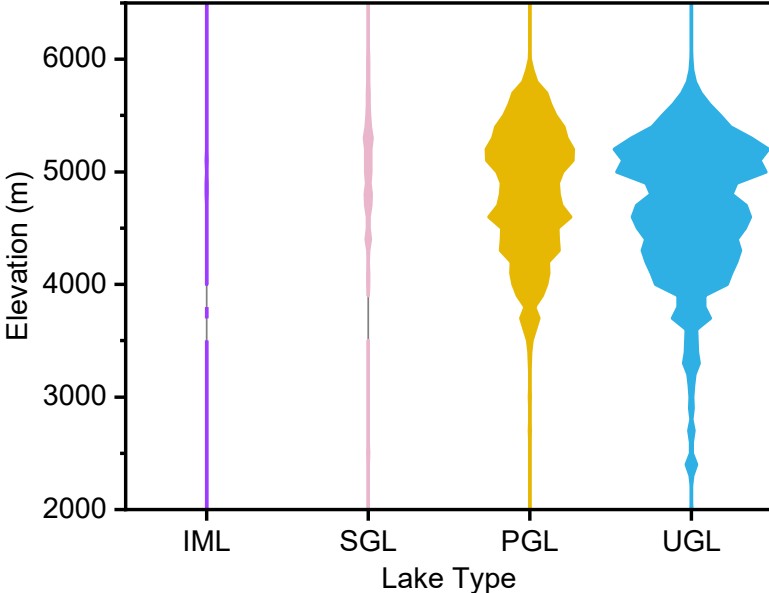

**Figure 7. Altitudinal characteristics of various types of glacial lake areas in 2013-2019.**

The number and area of glacial lakes for the three study periods were analyzed in different administrative regions (Figure 8 and Table 1). Figures 9-11 show the lake types, area change rate on 1°×1° grids. And Table 1 lists the tabulated data of glacial lake changes. The results show that Western Pamir and Eastern Hindu Kush presented a noticeable negative change in glacial lake areas, with a decrease of 2.937 km$^2$ and 8.651 km$^{2,}$ respectively. By contrast, the total area of glacial lakes in the Western Kunlun Mountains, Eastern Kunlun Mountains, and Tibetan Interior Mountains increased significantly, owing to the retreating and thinning of the debris-covered glaciers (Chen et al., 2021; Song et al., 2016). The increase of glacial lake areas was also observed in the Qilian Mountains, Central Himalaya, Nyainqêntanglha, and Hengduan Mountains. No significant change of glacial lakes was observed in the Karakoram.

In order to observe the changes of glacial lake areas in a more detailed manner, we took 1°×1° as a grid unit to make the analysis (see Figure 10). As can be seen from Figure 8 and Figure 10, in some mountain-wide regions, the number and area change trends of glacial lake areas in some places are contrary to the overall trend of the whole region (such as the left edge of the West Pamir and Qilian Mountains). Current studies targeting TP glacial lakes did not provide an in-depth analysis of interior TP (Chen et al., 2021; Wang et al., 2020; Zhang et al., 2020a) since there are fewer glaciers in the interior TP. Therefore, fewer glacial lakes are developed within 10 km of the glacial terminus compared to the entire TP, which does not have a dominant effect on the overall variability of all glacial lakes within the TP. In addition, the distribution percentages of the four types of glacial lakes were counted with the data of the third period (2013-2019) as an example (Figure 9). SGL was mainly distributed in the Nyainqêntanglha region, and a small amount was also distributed in the Himalayas. The West-South-Southeast zone of TP is composed of Western Pamir, Eastern Hindu Kush, Karakoram, Himalayas and Nyainqêntanglha. The area of PGL accounts for about half of all glacial lakes, and the trend is expanding. In the interior,





north and east of TP, UGL occupies the vast majority of glacial lakes. There are no large-scale glaciers in these areas, most of them are small independent glaciers, the interaction between glacial lakes and glaciers is weak (Chen et al., 2021), so climate factors such as precipitation are easier to affect the change of glacial lakes. Zhang et al. (2020b) also found that the Inner TP is getting wetter while southern TP is getting dryer.



**Figure 8. Area change rate of glacial lakes from the first period (1990-1999) to the last period (2013-2019) in various mountain-wide regions. The hillshade basemap is sourced from ESRI.**



**Figure 9. Proportional area distribution of four type glacial lakes in the last period (2013-2019) on 1°×1° grids. The circle size represents the total glacial lake area in the last period of each grid. The hillshade basemap is sourced from ESRI.**





**Figure 10. Area change rate of glacial lakes from the first period (1990-1999) to the last period (2013-2019) on 1°×1° grids. The circle size represents the total glacial lake area in the first period of each grid. The hillshade basemap is sourced from ESRI.**




**Table 1 Regional summary of lake numbers and areas in each period.**

| Region | Number | | | | Total area (km$^2$) | | | |
|---|---|---|---|---|---|---|---|---|
| | 1990-1999 | 2000-2012 | 2013-2019 | Change Rate (%) | 1990-1999 | 2000-2012 | 2013-2019 | Change Rate (%) |
| Altun Mountains | 25 | 18 | 18 | -28.00 | 0.429 | 0.511 | 0.447 | 4.20 |
| Central Himalaya | 2481 | 2619 | 2768 | 11.57 | 211.261 | 236.093 | 261.695 | 23.87 |
| Eastern Himalaya | 2583 | 2825 | 3160 | 22.34 | 205.757 | 216.980 | 245.662 | 19.39 |
| Eastern Hindu Kush | 1743 | 1795 | 1846 | 5.91 | 111.026 | 106.020 | 102.375 | -7.79 |
| Eastern Kunlun Mountains | 501 | 595 | 614 | 22.55 | 20.161 | 34.664 | 30.350 | 50.54 |
| Eastern Pamir | 101 | 102 | 104 | 2.97 | 14.404 | 20.587 | 15.764 | 9.44 |
| Eastern Tibetan Mountains | 268 | 252 | 306 | 14.18 | 18.496 | 20.751 | 21.384 | 15.61 |
| Gangdise Mountains | 1631 | 1770 | 1856 | 13.80 | 131.147 | 140.392 | 141.769 | 8.10 |
| Hengduan Mountains | 1596 | 1796 | 2108 | 32.08 | 93.484 | 98.435 | 112.163 | 19.98 |
| Karakoram | 243 | 209 | 229 | -5.76 | 23.903 | 21.435 | 24.656 | 3.15 |
| Nyainqêntanglha | 3879 | 4016 | 4788 | 23.43 | 322.247 | 325.930 | 396.054 | 22.90 |
| Qilian Mountains | 131 | 172 | 167 | 27.48 | 7.951 | 11.502 | 11.686 | 46.98 |
| Tanggula Mountains | 1348 | 1346 | 1368 | 1.48 | 59.669 | 72.740 | 69.920 | 17.18 |
| Tibetan Interior Mountains | 616 | 946 | 884 | 43.51 | 45.745 | 74.688 | 69.626 | 52.20 |
| Western Himalaya | 1108 | 1275 | 1263 | 13.99 | 103.494 | 99.547 | 110.458 | 6.73 |
| Western Kunlun Mountains | 171 | 208 | 203 | 18.71 | 27.373 | 45.100 | 44.295 | 61.82 |
| Western Pamir | 758 | 711 | 786 | 3.69 | 112.628 | 111.629 | 109.691 | -2.61 |



## 5 Discussion

### 5.1 Comparison with other glacial lake datasets

The deficiencies of state-of-the-art glacial lake inventories of the TP lie in the limited spatial and temporal resolution. Some inventories only cover small regions with significant missing data (Zhang et al., 2018a), others have only one to two periods of mapping data (Wang et al., 2020; Zhang et al., 2015), or cover only recent periods (Chen et al., 2021). However, limited by the quality and quantity of early satellite images, it is difficult to conduct year-by-year mapping of the whole TP glacial

lakes. In this regard, we adopted an image aggregation technique that has not been applied in the field of glacial lake mapping on the whole TP. Multi-year images with no or few clouds and no terrain shading were selected and fused into a single image under relevant parameters within the Google Earth Engine platform, and glacial lake boundaries were mapped. However, the differences in spatial and temporal coverage, as well as the different minimum threshold areas used in lake mapping also brings difficulties to a comparative evaluation. Because of these limitations, we compared only those studies

having a glacier area larger or equal to 0.0081 km$^2$ for the comparison analysis. With the same or larger study area (TP or HMA), equal to the minimum threshold area (0.0081 km$^2$), and definition of the location of a glacial lake (with the distance of 10 km from the nearest glacier terminus), we found that the glacial inventories of Chen et al. (2021) and Wang et al. (2020) are analogous and hence used to make a comparative analysis with our inventory.

Within the same minimum threshold area and spatial region, glacial lakes number and total area are statistically analyzed. As

shown in Table 2, there are noticeable differences among the three datasets. The inventory from Chen et al. (2021) has the least number and smallest total area, whereas the inventory from Wang et al. (2020) is closer to this study.

Since Chen et al. (2021) focused on short-term variability in recent years, their inventory is annual-based covering the time period from 2008 to 2017, which was greatly constrained by the image quality of the analyzed years. Hence, many lakes are omitted, which may be the main reason for the significant difference between the two inventories. Although Wang et al.

(2020) mapped more glacial lakes compared to Chen et al. (2021), but they only mapped two separate periods of data for 1990 and 2018, leaving considerable data gaps. To further compare, we put the three data inventories together for examination and randomly selected the more prominent parts of the regions of discrepancy for double-checking. According to Figures S1-S3 (Supplement), Wang et al. (2020)'s inventory does not cover well in some regions and omits a considerable number of glacial lakes, and Figures S4-S5 (Supplement) show the omission of Chen et al. (2021)'s data in some regions.

These comparisons prove that the proposed research can better fill the gaps in Chen et al. (2021)'s inventory in terms of temporal coverage, and can also better fill the gaps in Wang et al. (2020)'s regional coverage.

After carefully examined of these three inventories, five reasons that may lead to significant differences were found: (i) To obtain a more accurate distribution range of glacial lakes, the GAMDAM glacier inventory with higher quality was selected in our study to create the buffer of 10 km of distance from the glacier terminus to lakes (He and Zhou, 2022; Nuimura et al.,

2015; Sakai, 2019), while the other two datasets applied RGI and other glacier inventories. Different glacier inventories are



bound to create a difference in buffers, which leads to different numbers in the glacial lake datasets; (ii) due to the limited quality of satellite imagery in the early stages, we categorized time-series satellite images into three periods to extract the glacial lake boundaries, respectively. However, the other two datasets took images from each year or the closest year to the given one; (iii) there are differences in the acquisition month on selection of imagery. We chose July to November as a
conservative range, but Wang et al. (2020) chose a loose time of June to November; (iv) we applied more accurate terrain data of AW3D30 to avoid the influence of mountain shadow and some glaciers. On the other hand, Chen et al. (2021) and Wang et al. (2020) chose SRTM DEM, which has comparatively lower in accuracy in steep mountains (e.g., Avtar et al., 2015; Purinton and Bookhagen, 2017); (v) the difference caused by cross-validation and manual correction. Each interpreter will inevitably have subjective differences in their understanding of glacial lakes, which also leads to differences in results.

**Table 2 Number and area of glacial lakes in different datasets**

| Dataset sources | Time | Number | Area (km$^2$) |
|---|---|---|---|
| This study | 1990-1999 | 19183 | 1509.17 |
| | 2000-2012 | 20655 | 1637.01 |
| | 2013-2019 | 22468 | 1767.99 |
| Wang et al. (2020) | 1990 | 18025 | 1349.214 |
| | 2018 | 20250 | 1579.009 |
| Chen et al. (2021) | 2008 | 11149 | 1199.478 |
| | 2009 | 11572 | 1149.932 |
| | 2010 | 11590 | 1218.32 |
| | 2011 | 11712 | 1212.695 |
| | 2012 | 11758 | 1194.173 |
| | 2013 | 12473 | 1222.587 |
| | 2014 | 12385 | 1238.371 |
| | 2015 | 13356 | 1267.209 |
| | 2016 | 13073 | 1260.805 |
| | 2017 | 13601 | 1273.41 |

Considering the difference in time coverage of the three datasets, we selected the most recent inventory from the three datasets, i.e., (2013-2019) of our dataset, 2018 of Wang et al. (2020) and 2017 of Chen et al. (2021) to conduct the correlation analysis on spatial distributions. To make statistics clear, we aggregated the total glacial lake areas in TP on the 0.1°×0.1° grids, then conducted correlation analysis. As shown in Figure 11, the inventory of this study is significantly
correlated with the datasets of Wang et al. (2020) and Chen et al. (2021). The distribution of most points is close, and two curves with high correlation are fitted. Combined with the comparison made by Chen et al. (2021), it is proved that there is an excellent consistency among the three sets of data.

Above all, compared with other glacial lake inventories, our inventory covered a long temporal range, maps, and counts most glacial lakes in TP with a reasonable uncertainty to the maximum extent possible and proved highly correlated with other
datasets.




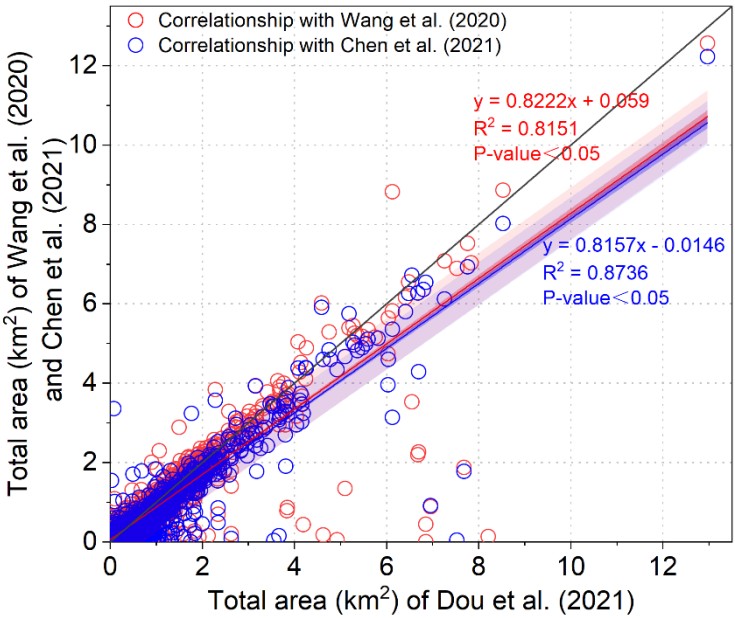

**Figure 11. Total area correlationship of our inventory with Wang et al. (2020) and Chen et al. (2021), respectively. The red and blue lines are the correlation fitting curves with each of them, the black line is the 1:1 diagonal line.**

## 5.2 Limitations and perspectives

In the process of this study, there were some known but not fully resolved problems and limitations that need to be acknowledged. Firstly, restricted by the spatial resolution of satellite images, not all glacial lakes of all sizes were mapped, parts of glacial lakes with an area of less than 0.0081 km² were excluded. Next, although we applied a fusion image approach to maximize the quality of the images, there were still some unavoidable clouds or mountain shadows ignored. As a result, a portion of the glacial lake would be missed. Meanwhile, due to the occurrence of extreme weather, some lakes

were covered by snow or ice floes, making it difficult to map the boundaries of all the glacial lakes. As more satellites with high-resolution and high revisit capabilities are launched and more powerful cloud computing platforms are established, it is possible to extend our datasets better and track the change of glacial lakes in TP for mapping and monitoring resources as well as associated hazards.

## 6 Conclusion

Integrating Landsat remote sensing images with GEE cloud-computing power, a detailed glacial lake inventory of the whole TP was mapped. The ID, area, length, mountain-wide range, and river basin of the glacial lakes were recorded in the attribute table of the dataset. Uncertainty analysis for glacial lakes shows that the average uncertainty for the whole region is

 

about 17.5%. The inventory has a high degree of consistency with other published works through the correlation analysis, which thoroughly verifies its reliability and scientificity.

We mapped a total of 22468 glacial lakes during 2013-2019 with the area of 1767.99 km², which makes our inventory the largest known dataset of glacial lakes in TP. Compared with the first period (1990-1999), the number of glacial lakes increased by 3285 (17.12%) and the area increased by 258.82 km² (17.15%). Glacial lakes are distributed unevenly in all the 17 mountains of TP, and the change rate of the area is different in each subregion. The elevation distribution of the glacial lake is analyzed with an interval of 100 m, and it is found that glacial lakes are mainly distributed in the range of 4400~5400

m a.s.l., with an evident expansion trend in recent decades. As glaciers retreat and climate change, the expansion of glacial lakes is still ongoing, especially for UGL and PGL. This freely available dataset will provide primary glacial lake data for all researchers interested in TP and support the study of climate change-glacier-glacial lake-GLOF interactions and hydro-climate models throughout the cryosphere.

## 7 Data availability

The Tibetan Plateau Glacial Lake Inventory (TPGL) is distributed under the Creative Commons Attribution 4.0 License. The data can be accessed from the data repository Zenodo at https://doi.org/10.5281/zenodo.5574289 (Dou et al., 2021).

## Abbreviations

a.s.l.: above sea level;

CGI: First and Second Chinese Glacier Inventory;

CPEC: China-Pakistan Economic Corridor;

DEM: digital elevation model;

GAMDAM: Glacier Area Mapping for Discharge from the Asian Mountains;

GLIMS: Global Land Ice Measurements from Space;

GLOFs: glacial lake outburst floods;

IML: Ice-marginal lakes;

MNDWI: modified normalized difference water index;

NDWI: normalized difference water index;

PGL: Proglacial lakes;

RGI: Randolph Glacier Inventory;

SGL: Supraglacial lakes;

SR: Surface Reflectance;

TP: Tibetan Plateau;



TPGL: Tibetan Plateau Glacial Lake database;

UGL: Unconnected glacial lakes.

**Author contribution**

**XD:** Conceptualization, Methodology, Programming, Formal analysis, Writing - Original Draft, **XF:** Conceptualization, Writing - Original Draft, Supervision, Project administration and Funding acquisition, **APY**: Methodology, Programming, Writing - Original Draft, **JX**: Formal analysis, Data Curation, Writing - Review & Editing, **RT**: Validation, Writing - Review & Editing, **XW**: Validation, Writing - Review & Editing, **QX:** Writing - Review & Editing, Supervision, Project administration and Funding acquisition.

**Competing interests**

The authors declare that they have no conflict of interest.

**Acknowledgment**

The authors would like to thank the following personnel for their participation in manual vectorization and correction: Lan Chen, Chengyong Fang, Liyang Jiang, Shikang Liu, Xinxin Tao, Zehao Xu, and Yinshuang Yang.

**Financial support**

This research is financially supported by the National Science Fund for Distinguished Young Scholars of China (Grant No. 42125702), and the Fund of SKLGP (SKLGP2019Z002).

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
