# Peer review of "Spatio-temporal evolution of glacial lakes in the Tibetan Plateau over the past 30 years"

_EGUsphere, 2022_

## Referee Comment (RC2)

**Review of '*Spatio-temporal evolution of glacial lakes in the Tibetan Plateau over the past 30 years*'**
**by Dou and co-authors, 2022**

The study by Dou and co-authors presents a new dataset of glacial lakes across all of High Mountain Asia (HMA) for three different study periods (1990-2019), mapped using Landsat 5 and Landsat 8 data in Google Earth Engine. The study highlights a general increase of glacial lake number and area, except for the most Western ranges of HMA.

This is a topical line of research, which tackles a very relevant question for the monitoring of hazards related to growing glacial lakes. However, I have a number of (very) major comments related to the soundness of the methods and results analysis, as well as to the novelty and relevance of the findings, that would need to be addressed for further consideration of this work.

I have also included a number of minor comments that can hopefully help the authors in revising their manuscript. These are however far from complete given the considerable revisions needed.

**Major comments**

**Introduction:** I do not find the discussion about the minimum lake threshold to be of much relevance in the introduction. The corresponding paragraph could be shortened considerably. Especially as the minimum size is not the only parameter that needs to be considered for the mapping – a (much) more important one would be the NDWI threshold (McFeeters et al., 1998), or even simply the method used for the mapping. The minimum lake threshold will indeed depend on the resolution of the images considered and is a basic threshold to post-process the mapping.

**Study focus:** The main aim of this study seems to have been the production of this glacial lake dataset for 3 distinct time periods. As such, the manuscript currently reads like a 'data' paper, with no clear research question or objective (the methods themselves as I understand from their limited descriptions, are also fairly basic). It is mentioned that there are some 'problems' related to lakes but it is unclear what these are and what this study does to address them.

**Methods:** The methods are very vague, preventing any assessment of their soundness. For example:

- There is only one sentence about the validation of the mapping with NDWI/MNDWI, with no quantitative information.
- The threshold selection is very obscure.
- Another critical aspect would be to specify how the aggregation was made for each period. Indeed, some lakes (the supraglacial ones especially) vary considerably in time (even seasonally). It is not clear how this is accounted for.

**Discussion:**

The discussion consists of a comparison of the dataset with other studies and considerations on the uncertainties. Interpretations (backed up with a sound statistical analysis) of the actual evolution patterns are clearly missing, and the added-value from this new inventory is not even mentioned.

**Figures:** In general not very informative and a few of them of poor quality.

**Minor comments**

**Title & Abstract**

L10: I do not think that the 'Tibetan Plateau' is the correct terminology here. This study spans most mountain ranges of High Mountain Asia (except the Tien Shan and the Altaï), including the Tibetan Plateau, which is here referred to as 'Tibetan Interior Mountains' (Fig. 1). In the current scientific consensus, the terminology in the Title and in all following text should therefore be 'High Mountain Asia (HMA)'.

L16: It would be useful to also indicate the relative values

**Introduction**

L29: references missing to support this statement.

L31: The term 'glacical lake' should be explicitly defined in the introduction.

L36: revise 'great important'

L40-43: references missing

L41-42: 'of computers' not necessary (repetition)

L55-60: A key missing reference here would be the work by Shugar et al., 2020

L57: Also Linear Spectral Unmixing (Kneib et al., 2020; Racoviteanu et al., 2021)

L79-80: last part of the sentence could be removed (self-explanatory)

**Study area**

L89-95: These are common facts, not particularly relevant for this study. Here it would on the contrary be useful to mention which area this study covers.

L96-100: So what? How is this related to this particular study?

L101-106: Here again, the link with the topic of the study is missing, and the text is very vague – it would need to be complemented with actual values.

**Data and methods**

L121: what do you mean by 'mainly'? Which images were used other than L5 TM and L8 OLI?

L124: In my experience a lot of ponds/lakes are already frozen in many regions of HMA in October/November. Can this be accounted for in the uncertainty estimation?

L124-125: I don't understand what you want to say here?

L132: what are the actual values? Here the authors are just mentioning a range?

L134: How was this used to remove shadows?

L140: In all the methods there is an approximation made between glacier and glacier terminus. It would be useful to better explain when either are used, and how the glacier terminus is defined. Here for example, it is not clear why the glacier terminus is used instead of the whole glacier outlines.

L144: glacier terminus or glacier?

L146: specify RGI version used – it would make sense to use different versions depending on the period considered.

L151: what is meant by 'updated time'?

L157: this is very vague – how many regions were selected, how large were they? A quantitative evaluation of this comparison would be required here, as well as for the choice of glacier outlines.

L160: The spatial and spectral properties of the 2 different sensors need to be actually compared (perhaps in a table?). Which products are used? Were they atmospherically corrected?

L161: English needs to be corrected.

L164: Is this implying that the threshold was chosen manually for every single of the ~43000 Landsat scenes? I find it hard to believe. How extensive are these checks.

L167: Either missing reference or missing explanation about these 'multiple attempts in different regions of TP'

L173-175: How is the 'front of the glacier' actually defined? What are the parameters used for the classification?

L176-177: How do you deal with the large interannual/seasonal variability (e.g. Miles et al., 2017) of these lakes? What does the value for each study period correspond to?

L179: This statement does not appear to be based on any scientific fact (the references mentioned do not refer to the Tibetan Plateau). Furthermore, how were these lakes differentiated from the proglacial lakes in this study?

L190: References? How many interpreters worked on this?

L196: based on the formula Aer is not proportional to the sensor resolution.

**Results**

L202: How were the lake extents aggregated from these 40000+ images?

L205: there should be the same number of significant digits for the mean and STD

L208: Why the shape? Where does it appear in Eq. 2?

L210: already stated in the methods.

L216: Are the lake acronyms really necessary? I feel that giving the names would simplify the text.

L216: What about the other types?

L219-220: what is the difference between an apparent and sharp increase? Actual values would be welcome here. Also for what follows.

L224: I had understood that these lakes had been removed? If yes, then they should not be considered in the results?

L225: This part belongs to the discussion. I anyways do not see how the authors get to this conclusion. A more thorough demonstration would be highly welcome.

L228: Missing reference to figure.

L231: Maybe the relative changes are, but one cannot see anything in figure 7.

L241: Administrative?

L244: significant digits.

L245: How is this significant?

L246: Move to discussion.

L248: Can you give actual values?

L251: left -> west

L263-264: Discussion.

**Discussion**

L281-282: This should be in the methods. More details about this aggregation would be welcome.

L286: check English.

L291: If the geographical extents of the different inventories are not the same, how did you account for this?

L295: remove 'but'

L300: This is not a 'proof'. These examples could be biased?

L302: English - examining.

L305: Vague. What 'other inventories'?

L308-309: Does aggregating over a full 10-year period really improve the mapping? Is there not on the contrary a risk to remove all the interannual variability?

L310: these time periods are actually really similar.

L314: It could make sense to make a comparison with the Shugar et al. (2020) dataset here.

L320: There is a correlation, but also a clear overestimation…

L322: Quite subjective statement. If the datasets are so consistent, what is the added-value of this new dataset?

**Figures**

Figure 1:

Why do the glacier lakes appear so big, covering entire mountains? Have they been buffered? I do not think it is the best way to represent them.

What is the source of the glacier outlines?

L109: Mention actual reference (Bolch et al., 2012)

L110: Mention resolution of basemap

Figure 3:

Missing scale, coordinates, lake and glacier outlines. In (b) the snow makes it hard to see anything. Why Google Earth instead of Landsat? This is not consistent with the study.

Figure 4:

(a) Should be in log scale. Ylabel of (b) is wrong.

Figure 5:

What about the other lake categories? Do (b) and (c) show all the lakes or only one category? This representation is likely biased by the hypsometry of the regions, it would be good to account for it one way or the other. It would also be interesting to compare the different regions.

Figure 6:

Missing region name, scale, lake and glacier outlines.

Figure 7:

I can't see anything for the IMLs and SGLs. What is the scale? It would be interesting to show the evolution here.

Figure 8:

Where does this rate come from? A linear trend? Would be good to show the actual values in each region. The term 'mountain-wide' does not make sense.

Figure 9:

Check English in the caption. I am somewhat skeptical about the relatively low area of supraglacial lakes, can you comment on this in your discussion?

Figure 10:

As for figure 8 the color scale is so large that it is hard to get a sense of the actual values. Suggest bounding it to 100%. As for figure 8, how is the rate of change calculated? Is the middle period not taken into account? What is the background? And the background outlines? Source?

Table 2:

Are the spatial extents of each dataset accounted for?

Figure 11:

'Correlationship' is not English.

---

## Referee Comment (RC3)

Dou et al. present a new dataset of lake area proximal (<10 km) to glaciers across High Mountain Asia which the authors suggest represents lake extents over three broad study periods (1990-1999, 2000-2012, 2013-2019). The study employs an image aggregation technique in Google Earth Engine to compile scenes from the Landsat 5 and Landsat 8 archives to increase image coverage over parts of the high mountain region which may otherwise have been obscured by clouds or shadows in individual scenes examined by previous inventory studies. The authors are able to map a greater number of lakes as a result.

Whilst the scope and approach of the paper seem appropriate, I don't believe the description of the methods employed by the authors are currently detailed enough to provide the reader with reassuring information about their efficacy. The aggregation of imagery from the three broad study periods is only addressed by the authors at the beginning of the discussion section, which is a significant oversight as the effectiveness of the technique is key to the rest of the paper in my opinion. The authors need to consider more carefully how this technique biases the effective date of composite imagery. The reader is currently given no information about when lake area observations from the three broad periods may have been from, which limits the conclusions which can be drawn about area change rates.

I also agree with previous reviewer of the paper that the analyses and resulting discussion of the new inventory is not far-reaching enough. The consistency of the area estimates have been compared with two existing inventories, but the authors make no attempt to examine the relationship between their inventories and other factors, such as glacier mass balance or various meteorological parameters, all of which are readily available. As a result, I don't think substantial new findings are apparent unfortunately. I would encourage the authors to expand the analyses of their inventory in this regard.

There is certainly potential for the study to yield interesting new findings, but substantial further work is still required at this stage, in my opinion.

I found a variety of more minor inconsistencies within the manuscript, which I list below:

L13- This statement gives the impression that an inventory is lacking, but this is not the case.

L40- Would be good to cite some early work on this topic here.

L43- Delete 'of computers' here.

L44- Landsat imagery generally has moderate resolution (at best 15 m). It can't be considered high resolution alongside WorldView, for example.

L61-73- This discussion of minimum lake threshold is rather repetitive and could be shortened considerably. To me the important point here is the later statement about the general lake area not changing much with different minimum thresholds (L77).

L81-82. This first sentence is contradictory. One says that most studies are sub-regional, the next mentions region-wide examples.

L85- I agree with the other reviewers here that, aside from the aim to produce a lake inventory, there are no clear research questions as such. A new dataset is helpful for sure, but there is much more that could be done with the new data.

Study area section- The setting of the geographical context isn't really required in its current fashion. It'd be more useful to describe the extent/coverage of the new inventory.

L114- 'mighty' not often used, suggest deleting this.

L116-117- How much of the work was manual in the process of generating the inventory?

L132- It's not clear what this approach actually does to remove cloud/shadows from imagery. A clearer explanation of this method is needed here, rather than just stating which tools were used in GEE.

L135- The use of a composite DEM is a strange choice. This DEM represents a surface which could be from any point between 2006-2011. It's not really clear how important this is to the authors method- can they offer more information on it's use? The following sentence about 'manual correction' is not clear.

L145-152- This could be considerably shortened to state that you used GAMDAM because it offers a contemporary record of glacier extent. The excessive citations in the lines above aren't needed.

L155-159- The authors need to provide some clear information about the results of this test case. Without a quantitative evaluation of these results it neither conclusive nor reassuring.

L163- For all images?! This sounds incredibly time consuming and subjective.

L165- Please provide the reader with a description of what these ENVI functions do.

L174- *All* proglacial lakes should be connected to a glacier, by definition (and described as such by Carrivick and Tweed, 2013).

L183- I'm not sure how a lake can be attributed as 'glacial' if it is not fed by a glacier? Fig.3 panel D shows this point- these lakes could be anywhere on the Tibetan Plateau, a long way from glaciers?

L206- But the image resolution is the same (30 m) without pan-sharpening. How is Landsat 8 just better?

L209- Repeated from earlier, would remove.

L221- What is 'compated'? Compared?

L225- This sounds very contradictory to the idea that glaciers commonly retreat (apart from those with substantial debris-cover) from their lowest elevation. Please rework this sentence.

Figure 6- I can only see the relation of one of these lakes to and nearby glacier (b) so the context of their expansion/contraction is missing. The reader isn't provided with any information about lake type here, so the comparison is a little meaningless.

Figure 7- What is plotted here? Lake area Vs altitude, lake number?

L245- The Kunlun Shan is part of the so called 'Karakoram anomaly', where glacier mass balance has recently been ~0, so I would revisit the idea that these glaciers are retreating and forming lots of new meltwater lakes. Similarly, debris-covered glaciers don't 'retreat' as quickly as clean-ice glaciers if their debris mantle is thick.

L257- Numerous studies based on higher resolution data have shown the prevalence and expansion of supraglacial ponds across debris-covered, Himalayan glaciers recently. The observation about there being few SGLs in the central Himalaya is related to data quality here and should be omitted.

~L260- There's a lot of interpretation mixed in with the reporting of results here, which should be saved for the discussion (which I note other reviewers have encouraged too).

Figures 8 and 9- The different approach to these two Figures really highlights an unfortunate contrast. Figure 8 suggests the strongest expansion in lake area in the Tibetan Interior Mountains, which to an inexperienced reader would suggest a large lake population here. Figure 9 contests this idea though, and actually (correctly) shows that there are very few lakes here. I think Figure 8 should be removed as a result.

L276-281- Much more on the aggregation technique needs to be included in the methods section and assessed in the results. The validity of the temporal analyses depends on this method and its description and it is currently not addressed.

L305- I doubt a different in glacier inventory extent of a few pixels (which would be the extent of the buffer contrasts) would result in the difference of >1000 lakes between inventories.

L310- The ALOS DEM has its limitations too- poor surface contrast, shadows and clouds all hinder the success of DEM extraction which do not effect the SRTM DEM. The authors are splitting hairs here which I don't think have had a big impact on results.